# Theoretical Evaluation of Potential Cytotoxicity of Graphene Quantum Dot to Adsorbed DNA

**DOI:** 10.3390/ma15217435

**Published:** 2022-10-23

**Authors:** Lijun Liang, Xin Shen, Mengdi Zhou, Yijian Chen, Xudong Lu, Li Zhang, Wei Wang, Jia-Wei Shen

**Affiliations:** 1Center for X-Mechanics, Key Laboratory of Soft Machines and Smart Devices of Zhejiang Province, School of Aeronautics and Astronautics, Zhejiang University, Hangzhou 310027, China; 2College of Automation, Hangzhou Dianzi University, Hangzhou 310018, China; 3School of Pharmacy, Hangzhou Normal University, Hangzhou 311121, China; 4Key Laboratory of Elemene Class Anti-Cancer Chinese Medicines, Engineering Laboratory of Development and Application of Traditional Chinese Medicines, Collaborative Innovation Center of Traditional Chinese Medicines of Zhejiang Province, Hangzhou Normal University, Hangzhou 311121, China; 5Key Laboratory of Surface & Interface Science of Polymer Materials of Zhejiang Province, Department of Chemistry, Zhejiang Sci-Tech University, Hangzhou 310018, China; 6Department of Pharmacy, Hangzhou Third People’s Hospital, Affiliated Hangzhou Dermatology Hospital, Zhejiang University School of Medicine, West Lake Road 38, Hangzhou 310009, China

**Keywords:** DNA, graphene quantum dot, cytotoxicity, molecular dynamics simulation, adsorption

## Abstract

As a zero-dimensional (0D) nanomaterial, graphene quantum dot (GQD) has a unique physical structure and electrochemical properties, which has been widely used in biomedical fields, such as bioimaging, biosensor, drug delivery, etc. Its biological safety and potential cytotoxicity to human and animal cells have become a growing concern in recent years. In particular, the potential DNA structure damage caused by GQD is of great importance but still obscure. In this study, molecular dynamics (MD) simulation was used to investigate the adsorption behavior and the structural changes of single-stranded (ssDNA) and double-stranded DNA (dsDNA) on the surfaces of GQDs with different sizes and oxidation. Our results showed that ssDNA can strongly adsorb and lay flat on the surface of GQDs and graphene oxide quantum dots (GOQDs), whereas dsDNA was preferentially oriented vertically on both surfaces. With the increase of GQDs size, more structural change of adsorbed ssDNA and dsDNA could be found, while the size effect of GOQD on the structure of ssDNA and dsDNA is not significant. These findings may help to improve the understanding of GQD biocompatibility and potential applications of GQD in the biomedical field.

## 1. Introduction

Due to its unique electrical and mechanical properties, graphene has been extensively studied in the past few years [1,2,3]. For example, graphene-based material is a strong absorber of lactate molecule, this feature can be very useful in designing highly sensitive sensors [4]. Graphene sheets can also be used to synthesize fast responding TSH nano-biosensors [5]. Graphene quantum dots (GQDs) are originated from graphene and graphene oxide, and are a new type of carbon nanomaterial formed when the two-dimensional scale of graphene sheets is restricted to a very small area and thus has quantum properties [6,7]. As zero-dimensional carbon-based nanomaterials, GQDs not only inherit the unique physical, chemical and mechanical properties of 2D graphene but also possess outstanding optical properties [8,9,10], excellent biocompatibility [11,12,13], as well as tunable surface functionalities [14,15], bringing a wide range of promising applications in bioimaging [16,17], biosensing [18,19,20], drug delivery [21], antibacterial [22], and cancer therapy [23,24,25]. Gong et al. [26] pointed out that GQDs could serve as excellent bioimaging agents. Xue et al. [27] suggested that GQDs with appropriate size may assist in the drug delivery process by reducing the translocation free energy permeating into the biomembrane. Yew et al. [28] reported that the GQDs could be used as fluorescence nanoquenchers for DNA detection. In addition, the modified GQDs also have a wide range of applications. Agrawal et al. [29] found that the modified GQDs have efficient modification as well as a higher stability, and enhanced biosensing efficiency and antibiofilm ability. Iannazzo et al. [30] pointed out the GQDs-18-crown-6 composite exhibited ratiometric fluorescence emission behavior with the variation of K+ concentration, demonstrating its promising properties for the development of a selective fluorescent method for potassium determination. Ajgaonkar et al. [31] developed a low-cost biocompatible and nitrogen-doped graphene quantum dots (NGQDs) sensor for pancreatic cancer miRNA (miRNA-132). This detection capability warrants the potential for ex vivo cancer miRNA detection with the advantages of being low-cost, simple, and noninvasive.

DNA is a pair of polynucleotides coiled around a common central axis. It is a part of chromosomes that exist in the nucleus and have the function of storing genetic information. DNA is a high-molecular polymer. Higher temperature, organic solvent, acid-base reagent, urea, amide, etc. can cause the denaturation of a DNA molecule, that is, the hydrogen bond between DNA double-stranded bases are broken and the double helix structure is untied. The stability of the DNA structure is closely related to human health. Mutations in DNA molecules will cause neurodegenerative diseases, such as Huntington’s disease, Alzheimer’s disease, and Parkinson’s disease. DNA, being a genetic information carrier in living cells reveals tunable semiconducting response in the presence of external electric and magnetic fields, which is promising for molecular electronics. Khatir et al. [32] revealed that the gold-DNA-gold structure is a promising magnetic sensor. Studies have also found that the combination of nanomaterials and DNA can be used in the field of biomedicine, for example, DNA can be complexed with graphene oxide for the application of biosensors because of its good adsorption [33,34]. However, the influence of graphene and its derivative on the structure and potential toxicity of DNA cannot be ignored. By the mean of MD simulation, Zeng et al. [35] found that graphene and its oxides can damage the base pairs of DNA in the process of adsorption, affecting the structure of DNA and then may cause biological toxicity. Gu et al. [36] found that graphene can destroy the structure of double-stranded DNA base pairs and break the hydrogen bonds between double-stranded DNA base pairs (especially the hydrogen bonds between ATs) through experiments and MD simulation, thus affecting the biological function of DNA. At the same time, questions about the biosafety of GQD have also attracted much attention in recent years. In a previous study, we found that the cytotoxicity of GQD with a small size is relatively low to the cell membrane and may be appropriate for biomedical application [37]. We also found that only GQD with a small size could enter into the interior of the DNA fragment and break the hydrogen bonds of the DNA fragment [38]. The GQD with a larger size tends to aggregate into a cluster and adsorb on the DNA fragment. These results suggest that the genotoxicity of GQD is relatively low and may be appropriate for biomedical applications. Although we have summarized the effects of different factors, such as size, geometry, and degree of oxidation, etc. of GQD on the cytotoxicity, from both experimental and theoretical perspectives [39], it is still necessary to reveal the underlying mechanism of interaction between GQD and DNA. In particular, the effects of different sizes, oxidation and oxidation groups of GQD on the structure of DNA are still obscure, which limits the understanding of GQD cytotoxicity.

Besides experiments, molecular dynamics (MD) simulations have been extensively applied to investigate the interaction between biomolecules and nanomaterials in recent years [40,41,42]. For example, Zhao et al. [43] used MD simulation to study the interaction between DNA fragments and the surface of graphene in an aqueous solution and found that the structure of double-stranded DNA (dsDNA) would be destroyed during the adsorption process on the graphene surface. In previous studies, experiments have focused on the biocompatibility of GQDs and GOQDs. However, the atomic details of the interaction mechanism between DNA and GQDs have not been well understood. Several works focused on the adsorption and configuration changes of single-stranded (ssDNA) as well as dsDNA on GQDs, and they did not consider GOQDs with different sizes [38,44]. In this study, we used the standardized model to generate GOQDs with random distribution of oxidation groups of different sizes through home-based script; both ssDNA and dsDNA were selected as model DNA to study the effects of different sizes and the oxidation of GQD on the structure and function of DNA by using MD simulation.

## 2. Methods

### 2.1. System Setup

The structures of GQDs with different sizes were selected in this study, which is as same as our previous works [45,46]. One C atom is set as the central atom of Cartesian coordinates (0, 0, 0), and the other C atoms satisfying x^2^ + y^2^ < R^2^ are chosen to be the C atoms of GQDs, where R is the radius of GQDs. We define the model of GOQD as the Lerf–Klinowski model generated by the standard oxidation process [47,48,49], namely C10O1(OH)1(COOH)0.5. More specifically, with an average of twenty C atoms, there are two epoxy groups and two hydroxyl groups randomly distributed on the GQD surface, while one carboxyl group randomly linked at the edge of the GQD on both sides of the plane. In this study, the geometry of all GQDs and GOQDs are circular, and the dimensions of GQD and GOQD are defined according to the number of carbon rings, which are GQD19 (GOQD19), GQD61 (GOQD61) and GQD275 (GOQD275), respectively, as shown in Figure 1a. The GQD and GOQD structures were optimized by Gaussian 03 [50], and then the optimized GQD and GOQD structures are simulated as the initial structures.

Deoxynucleotides are composed of nitrogen-containing bases, deoxyribose and phosphate [51]. The ssDNA and dsDNA structures selected in this article are all constructed from HyperChem8.0. The ssDNA is structured in poly(C)_6_ containing six cytosine and poly(G)_6_ containing six guanine. The dsDNA consists of six repeated C-G, namely poly(CG)_6_, as depicted in Figure 1b. All ssDNA and dsDNA were placed with their central axis parallel to the surfaces of GQD and GOQD, and the shortest distance between DNA and GQD (GOQD) in each system is approximately 0.8 nm in all simulated systems. In order to neutralize the charge of the systems, 5 Na^+^ were added to the systems containing ssDNA, and 10 Na^+^ were added to the systems containing dsDNA.

### 2.2. MD Simulations

All MD simulations were performed by GROMACS version 5.0.4, and all simulations were based on the CHARMM36 force field. The CHARMM36 force field provides many parameters of biomolecules, such as proteins, lipids, nucleic acids, which is very suitable for simulation of DNA and nanomaterials [52]. The water molecule is represented by the TIP3P model [53], which is widely used in biological systems. The number of water molecules in each system is from 6039 to 12,047, based on the size of the simulated systems. The harmonic bond potentials of C-C and C-H bonds of GQD, the harmonic angles of C-C-H and C-C-C bonds, the potential parameters of the harmonic dihedral angles and the non-bonded Lennard–Jones (LJ) parameters were all derived from the work of Cohen-Tanugi et al. [54]. The bonded and non-bonded parameters of carboxyl, hydroxyl and epoxy were obtained from the B3LYP/6-311+G(d, p) level calculation in Gaussian 03 software [50] and AmberTool [55]. Moreover, all chemical bonds, including those linked to the H atoms, were constrained by the LINCS algorithm [56]. The algorithm is inherently stable, as the constraints themselves are reset instead of derivatives of the constraints, thereby eliminating drift. Although the derivation of the algorithm is presented in terms of matrices, no matrix multiplications are needed and only the nonzero matrix elements have to be stored, making the method useful for very large molecules. In all simulations, the v-rescale method was used to maintain the temperature at 310 K, and the pressure was kept at 1 bar by the Berendsen thermostat. The time step in all simulations was set 2 fs. The Lennard–Jones (LJ) interactions were treated with a cutoff distance of 1.2 nm, and the particle mesh Ewald (PME) method with a cutoff of 1.2 nm was used to calculate the electrostatic interactions. All systems undergone 50,000 steps of energy minimization and were relaxed for 1 ns under the NVT ensemble. The production runs of ssDNA adsorption on the surface of GQDs or GOQDs were carried out for 200 ns in NPT ensemble. Since the adsorption of dsDNA on the surface of GQDs or GOQDs is accompanied by the unwinding of dsDNA and the adsorption dynamics is relatively slow comparing to the adsorption of ssDNA, the production runs of dsDNA adsorption were extended to 500 ns in NPT ensemble. The simulation details of all systems were displayed in Table 1. The trajectories and snapshots of all systems were primarily visualized by VMD [57].

## 3. Results and Discussion

### 3.1. The Adsorption Behavior of DNA onto GQD and GOQD

ssDNA was found to adopt a flat orientation when adsorbed on the surface of GQD and GOQD after 200 ns MD simulation, as shown in Figure 2a. In order to explain the adsorption behavior of ssDNA in detail, the changes of angle (θ) between ssDNA and GQD as well as that between ssDNA and GOQD were calculated (Figure 2b). Herein, θ is defined as the angle between the axis vector of ssDNA and the normal vector of GQD (GOQD) plane. At the initial state of the simulation θ = 90° (cos θ = 0), indicating that the axis of ssDNA is parallel to GQD and GOQD. During the simulation, ssDNA quickly lay flat and adsorbed on the surface of GQD19 (cos θ = 0), while the angle between ssDNA and the surface of GOQD19 fluctuates relatively greatly, which may be due to the existence of hydrophilic groups on GOQD surface. The hydrophobic interaction between ssDNA and GOQD surface is relatively weaker so that the ssDNA cannot be rapidly adsorbed on the surface of GOQD. Compared with poly(C)_6_, the angle fluctuation between poly(G)_6_ and GQD as well as that between poly(G)_6_ and GOQD surface is smaller, attribute to the fact that guanine has larger conjugated structure and stronger π–π stacking interaction with GQD (GOQD), therefore, it can be adsorbed more firmly on the surfaces of GQD and GOQD.

With increased size of GQD or GOQD, ssDNA could still be adsorbed on their surface in a flat orientation, but θ in ssDNA-GOQD61 systems fluctuates more greatly. Compared with poly(C)_6_, the angle fluctuation between poly(G)_6_ and the surface of GQD and GOQD is smaller, as shown in Appendix A. With the largest size of GQD, the adsorption of ssDNA on the surface of GQD275 and GOQD275 is more obvious, as shown in Appendix A. In Appendix A, it can be observed that comparing with the smaller GQD and GOQD, ssDNA can be more quickly adsorbed on the surface of GQD275 and GOQD275 with flat orientation, and the angle fluctuation between ssDNA and GQD275/GOQD275 is the smallest. This may be attributed to the stronger hydrophobic interaction between ssDNA and GQD275/GOQD275, so that ssDNA can be adsorbed more quickly and firmly. In Appendix A, the angular fluctuation between ssDNA and the surface of GOQD275 is relatively obvious, especially for poly(G)_6_. Poly(G)_6_ is finally adsorbed at the edge of GOQD275 rather than in the middle of GOQD275, which led to a rapid change of molecular orientation of poly(G)_6_.

To gain more insight about the dynamics of the adsorption of ssDNA on GQD and GOQD surfaces with different sizes, the time evolution of the distance of center of mass (COM) between ssDNA and GQD/GOQD along the z-axis (perpendicular to GQD or GOQD surface) during the adsorption process was calculated, as shown in Figure 3. The results demonstrate that with larger size of GQD, the ssDNA adsorption on GQD surface is faster, and the COM distance is smaller. This result may be explained by stronger hydrophobic interaction and π–π stacking interaction between poly(C)_6_/poly(G)_6_ and GQD with more conjugated surface area. However, with the increase of GOQD size, the COM distance between poly(C)_6_/poly (G)_6_ and GOQD is not that obvious. The reason is probably that there are certain number of hydrophilic groups on the surface of GOQD, and these hydrophilic groups destroyed the conjugation structure of GQDs, so the π–π stacking effect and hydrophobic interaction between GOQD and poly(C)_6_/poly(G)_6_ are greatly weakened, which reduces the size effect of GOQD on the adsorption of two kinds of ssDNA.

Different from the adsorption orientation of ssDNA, we found that dsDNA tended to be adsorbed on the surface of GQD and GOQD in a vertical orientation after 500 ns MD simulation, as shown in Figure 4a,b. At the beginning of the simulation, the dsDNA was located on the surface of GQD19 and GOQD19 in parallel orientation. During the simulation, the dsDNA gradually breaks part of the helix structure and adjusted its adsorption orientation on the surface. Finally, the dsDNA was stably adsorbed on the surface of GQD19 and GOQD19 in a nearly vertical way until the end of the simulation. In order to explore this phenomenon in detail, the angle between the axis of two single strands of dsDNA and the normal of GQD19/GOQD19 surface were calculated. From the data of Figure 4c, one can find that dsDNA quickly adjusts its orientation on the GQD surface. One of the chains tends to move close to surface with the vertical orientation (cos θ = 1.0) and firm adsorption on GQD19 and GOQD19 surface, and the other chain also adopted a vertical adsorption orientation after relatively long simulation time. Therefore, the double strands of DNA adsorbed on the surface of GQD19 and GOQD19 in a nearly perpendicular way, which is consistent with the snapshot shown in Figure 4a,b. The difference between the adsorption behavior of ssDNA and dsDNA could attribute to the fact that the bases of ssDNA are directly exposed to the surface and can be rapidly adsorbed on the surfaces of GQD and GOQD, while the base pairs of dsDNA are wrapped inside and maintain the stability of the double helix structure relying on the π–π stacking interaction in the same chain and the hydrogen bonds between base pairs from different strand. During the process of adsorption, the dynamics of unwinding of the double helix is slow, with the competition between surface affinity and inter-chain interaction (π–π stacking and hydrogen bonding). Therefore, dsDNA interacts with GQD or GOQD through outermost two bases, while the double-stranded structure is partially broken and adsorbed on the surface of GQD and GOQD in a vertical orientation, but stabilizing the DNA-GQD/GOQD complex structure.

If the sizes of GQD and GOQD increases, one can still find that dsDNA is adsorbed on the surface of GQD61 and GOQD61 in a vertical orientation at the end of the simulation. As shown in Appendix A, due to the larger size of GQD61 and its enhanced attraction to dsDNA, the dynamics of unwinding of dsDNA is accelerated, resulting in one chain of dsDNA adsorbed on its surface in a nearly parallel manner at the end of the simulation. Appendix A displays the angle between the axis of two single strands of dsDNA and the normal of the GQD61/GOQD61 surface as a function of simulation time, and it confirm the results from the simulation snapshot. Compared with GQD61, GOQD61 has a weaker hydrophobic interaction with dsDNA due to its increased surface hydrophilicity, so one chain in dsDNA is always adsorbed on the surface of GOQD61 in a nearly vertical orientation in the later stage of simulation, while the other chain adsorbs to the surface of GOQD61 in a similar orientation after sharp fluctuations.

If the size of GQD is the largest, the adsorption of dsDNA on the surface of GQD275 is the strongest, as shown in Appendix A. Compared with GQD19 and GQD61, the adsorption of dsDNA on the surface of GQD275 is stronger, and the dynamics of unwinding is faster. One of the strands of dsDNA is adsorbed on the surface of GQD275 in a nearly vertical orientation, while the other chain is adsorbed on the surface of GQD275 in a nearly parallel orientation in a short time scale due to stronger π–π stacking interaction and hydrophobic interaction, as shown in Appendix A. On the surface of GOQD275, although the final adsorption orientation of dsDNA to the surface is similar to that of GQD275, the orientation fluctuation during the adsorption process is greater owing to the weakening of π–π stacking interaction and hydrophobic interaction.

It can be seen from Figure 5 that the COM distance between dsDNA and GQD/GOQD is larger than that between ssDNA and GQD/GOQD (see Figure 3). This is because the base pairs of ssDNA are exposed to the surface and can interact with GQD/GOQD directly, and they tend to be adsorbed on the surface of GQD/GOQD in a parallel manner. From Figure 5a, larger size of GQD possesses the faster dsDNA adsorption dynamics on its surface and the smaller COM distance between GQD and dsDNA. This result is as same as the adsorption of ssDNA on the GQD surface. This could be explained by the fact that with the increase of the GQD size, its hydrophobic interaction with the dsDNA and π–π interaction becomes stronger, making the dsDNA adsorbed on the surface of the GQD much faster. In Figure 5b, the COM distance between dsDNA and GOQD decrease with the increase of GOQD size, but the difference is not noticeable. This phenomenon is the same as the adsorption of ssDNA on GOQD surface (discussed above), owing to the decrease of π–π stacking effect and the weakening of hydrophobic interaction between dsDNA and GOQD.

Based on the above analysis and discussion, the adsorption of ssDNA on the surface of GQD and GOQD tends to be in a parallel manner. This stems from the fact that the exposed bases of ssDNA can directly form π–π stacking interaction and stronger hydrophobic interaction with surfaces. The unwinding process of dsDNA requires to overcome a certain energy barrier, and the dynamics is relatively slow. It is difficult for the bases fully exposed to the surface of GQD and GOQD, so they prefer to be adsorbed on the surface of GQD and GOQD in a nearly vertical way. These conclusions are consistent with the results of DNA adsorption orientation on the surface of graphene studied by Zeng [35] and Zhao et al. [43]. It is also consistent with our recent study on the adsorption orientation of DNA on the surface of two-dimensional material MoS_2_ [58].

### 3.2. Structural Evolution of ssDNA Adsorbed onto GQD and GOQD

To analyze the structural changes of ssDNA during the adsorption process, the RMSD of ssDNA and the number of contacts between ssDNA and GQD/GOQD were calculated, as shown in Figure 6. It is obvious that as the size of GQD increases, the RMSD value of poly(C)_6_ increases, and so does the number of contacts between poly(C)_6_ and GQD (see Figure 6c). As the increase of the size of GQD, the π–π interaction between GQD and poly(C)_6_ as well as the hydrophobic interaction increase, and hence the stronger adsorption of poly(C)_6_ on surface, therefore the structural change of poly(C)_6_ is more obvious.

However, for the dsDNA-GOQD systems, the hydrophobic interaction with poly(C)_6_ is relatively weak, and the RMSD value of poly(C)_6_ fluctuates more greatly (see Figure 6b). Compared with GQD systems, there are fewer contact atoms between GOQD and poly(C)_6_ (see Figure 6d). There are more solvents (adsorption of water molecules) on the surface of oxygen-containing functional groups on GOQD surface, which hinders the direct adsorption of poly(C)_6_ to some extent, which is consistent with the results of Zeng et al. [35]. We also found that in GOQD systems, as the GOQD size increases, the RMSD value of poly(C)_6_ does not differ significantly, and the number of contacts between poly(C)_6_ and GOQD did not increase with the increase of GOQD size. The electrostatic interaction between GOQD and ssDNA mainly through the hydrophilic hydroxyl and epoxy groups on the surface, and the hydrophobic interaction is relatively weak, this is reason that the size effect of GOQD on the adsorption of ssDNA is not as obvious as that of GQD.

The RMSD value of poly(G)_6_ and the number of contacts between poly(G)_6_ and GQD/GOQD were also calculated and displayed in Appendix A. If poly(G)_6_ adsorbed on the surface of GQD275, its RMSD value and the number of contacts is the largest among all GQDs, attribute to the largest π–π stacking effect and hydrophobic interaction between poly(G)_6_ and GQDs. Meanwhile, compared with poly(C)_6_, the guanine of poly(G)_6_ has larger conjugated structure in its base, and the π–π stacking and hydrophobic interaction between poly(G)_6_ and GQD is stronger than that between poly(C)_6_ and GQD, so the RMSD value of poly(G)_6_ is higher than that of poly(C)_6_ on GQD with the same size. We also found that the RMSD of poly(G)_6_ adsorbed on the GQD19 surface was higher than that on the larger GQD61 surface. The guanine at both ends of poly(G)_6_ were stably adsorbed on the GQD19 surface (see Figure 2a), lead to larger conformational change of the single strand.

For ssDNA-GOQD systems, the RMSD of poly(G)_6_ and the number of contacts also increase with the increase of GOQD size. It is clear that the guanine contains a larger conjugated structure, and its π–π stacking and hydrophobic interaction with the GOQD surface is stronger than that of the cytosine, which can still reflect a certain degree of size effect. This is consistent with Antony’s [59] finding that the order of π–π stacking interaction between bases and graphene is G > A > T > C. However, the number of contacts between poly(G)_6_ and GOQD275 does not significantly exceed the number of contacts between poly(G)_6_ and GOQD61.

### 3.3. Structural Evolution of dsDNA Adsorbed onto GQD and GOQD

To investigate the structural changes of dsDNA adsorbed on the surfaces of GQD/GOQD, the RMSD value of dsDNA and the number of contacts between dsDNA and GQD/GOQD were calculated. As shown in Appendix A, the RMSD value of dsDNA adsorbed on GQD surface and the number of contacts increases with the size of GQD increases. In Appendix A, the RMSD of dsDNA did not show a similar trend with the increase of GOQD size, and the RMSD fluctuated most when dsDNA was adsorbed on GOQD61. This may be caused by the strongest uncoiling of dsDNA double helix structure in this system. In Appendix A, the number of contacts between dsDNA and GQD/GOQD increases with the increase of GQD/GOQD size, however, the size effect of GQD is stronger than that of GOQDs.

To gain more information on the structural changes of dsDNA adsorbed on GQD/GOQD surface, the number of π–π stacking between the internal base pairs of dsDNA in each system during the simulation were calculated, as shown in Figure 7. Of three sizes of GQD/GOQD, in most cases, the number of π–π stacking slightly reduces with the simulation time increase, which reflects that the double helix structure of dsDNA is partially unwound during the process of adsorption in 500 ns MD simulation, owing to the competition between surface affinity and inter-chain interaction (π–π stacking and hydrogen bonding), as we discussed in Section 3.1. In the case of GOQD61-poly(CG)_6_ system, the number of π–π stacking of dsDNA reduced most, from 10 at the beginning of the simulation to 3 at the end of the 500 ns simulation, which indicate larger deformation of the structure, as we discussed above in Appendix A.

The change of the number of hydrogen bonds between the base pairs of dsDNA during the adsorption was also calculated. As shown in Figure 8a, the number of hydrogen bonds between dsDNA base pairs decreases with the increase of GQD size, indicating that larger size of GQD lead to more obvious damage to the dsDNA double helix structure, which is consistent with the RMSD evolution in Appendix A.

Moreover, the decrease of the number of hydrogen bonds in dsDNA-GQD system was larger than that in dsDNA-GOQD systems (except for GOQD61-poly(CG)_6_ system), which consist with the evolution of number of π–π stacking discussed in Figure 7.

Through MD simulation, Zeng et al. [35] found that graphene oxide has less structural damage and lower cytotoxicity to DNA than graphene, due to the stronger adsorption of bases in DNA on more hydrophobic graphene. In this study, we found that the adsorption of ssDNA and dsDNA on the surface of GQD has stronger size effect. Comparing to GOQD, the larger the size of GQD, the more obvious damage to DNA structure. In particular, GQD275 caused the most obvious damage to DNA structure. GQD275 not only lead to the most atomic contacts with poly(C)_6_, poly(G)_6_ and poly(CG)_6_ but also had the largest changes in RMSD values of them on its surface. Meanwhile, the destruction of ssDNA and dsDNA structure on GQD surface is higher than that on GOQD surface. These results are consistent with the results of Jeong et al. [44]. By means of experiments and MD simulation, Jeong et al. found that compared with GOQD, GQD has stronger adsorption to ssDNA and greater damage to the structure of ssDNA. In addition, Wang et al. [60] found that GQD can induce DNA damage in NIH-3T3 cells, which lead to cytotoxicity. Our results suggest that although GQD and GOQD have lower cytotoxicity than graphene, they may also induce the structure change of DNA to a certain extent, causing potential cytotoxicity. Moreover, GQD has greater influence on the structure change of DNA than GOQD, and the size effect of GQD on DNA structure change is more obvious than that of GOQD.

## 4. Conclusions

In this study, MD simulation was used to explore the adsorption behavior and dynamics of ssDNA/dsDNA on GQD and GOQD with different sizes. The effects of size and oxidation of GQD were investigated. In addition to investigating the interaction between DNA and GQD with different sizes, we also focused on the effect of GOQD with various sizes on the adsorption of ssDNA as well as dsDNA, which was not addressed in previous studies. During the adsorption process, ssDNA tends to adsorb on the surfaces of GQD/GOQD in a parallel manner, while dsDNA tends to adsorb on the surfaces of GQD/GOQD in a nearly perpendicular orientation. GQD has greater influence on the structure change of DNA than GOQD, and the size effect of GQD on DNA structure change is more obvious than that of GOQD. The larger the size of GQD, the more obvious the damage to the DNA structure. These results indicate that although GQD and GOQD have lower cytotoxicity than graphene, which is confirmed by recent experiment and simulation studies, they may also induce the structure change of DNA to a certain extent, causing potential cytotoxicity, and hence need to be investigated more deeply for their biosafety in future. In future work, we plan to study the effect of GOQD concentration, the degree of oxidation and DNA length on the interaction between GOQD and DNA. Moreover, the interaction mechanism between GQD/GOQD and protein, as well as the transmembrane process of GQD/GOQD are worth investigating to fully understand the cytotoxicity of GQD/GOQD materials.

## Figures and Tables

**Figure 1 materials-15-07435-f001:**
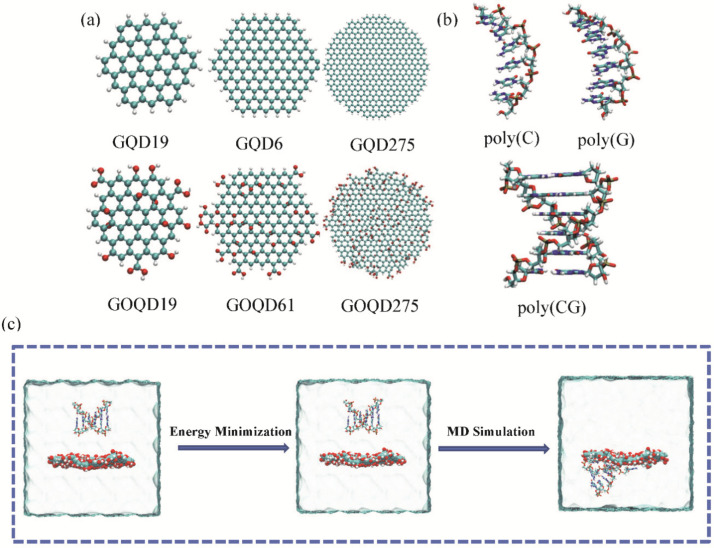
(**a**) Graphene quantum dots with different sizes and oxidation: GQD19, GQD61, GQD275, GOQD19, GOQD61, and GOQD275, respectively; (**b**) The initial structure of ssDNA and dsDNA used in this study. (**c**) Schematic diagram of the system from its initial conformation to conformations after energy minimization and MD simulation. Water is shown transparently.

**Figure 2 materials-15-07435-f002:**
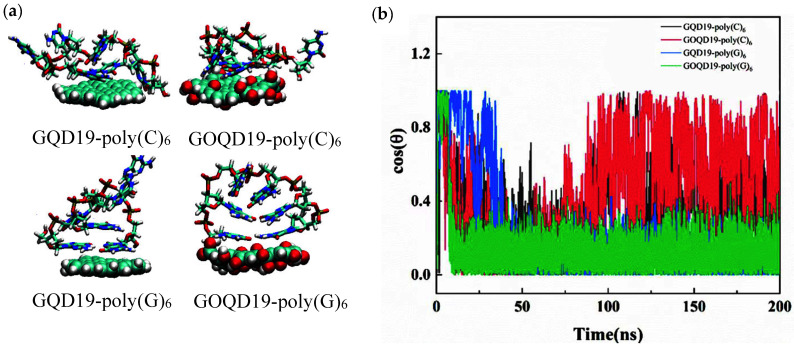
(**a**) Representative snapshots of ssDNA adsorption on GQD19 and GOQD19 surface at the end of the 200 ns MD simulation. (**b**) The curve of the angle between the ssDNA axis vector and the normal vector of the GQD19 (GOQD19) plane during the simulation.

**Figure 3 materials-15-07435-f003:**
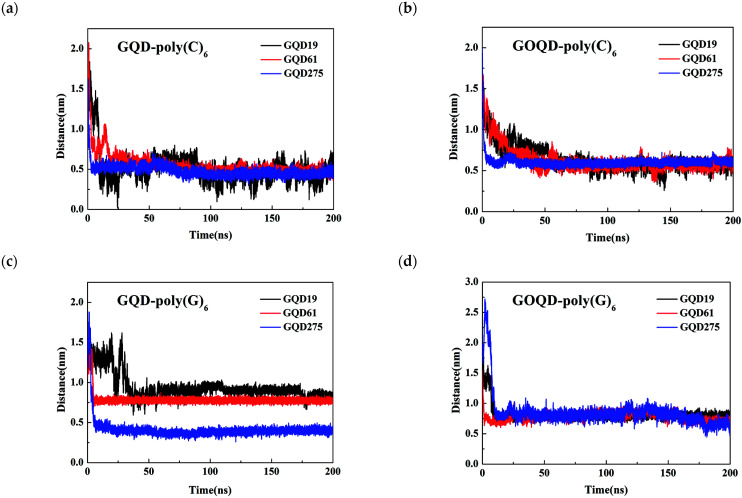
The COM distance of between ssDNA and GQD/GOQD during the adsorption process: (**a**) GQD-poly(C)_6_; (**b**) GOQD-poly(C)_6_; (**c**) GQD-poly(G)_6_; (**d**) GOQD-poly(G)_6_.

**Figure 4 materials-15-07435-f004:**
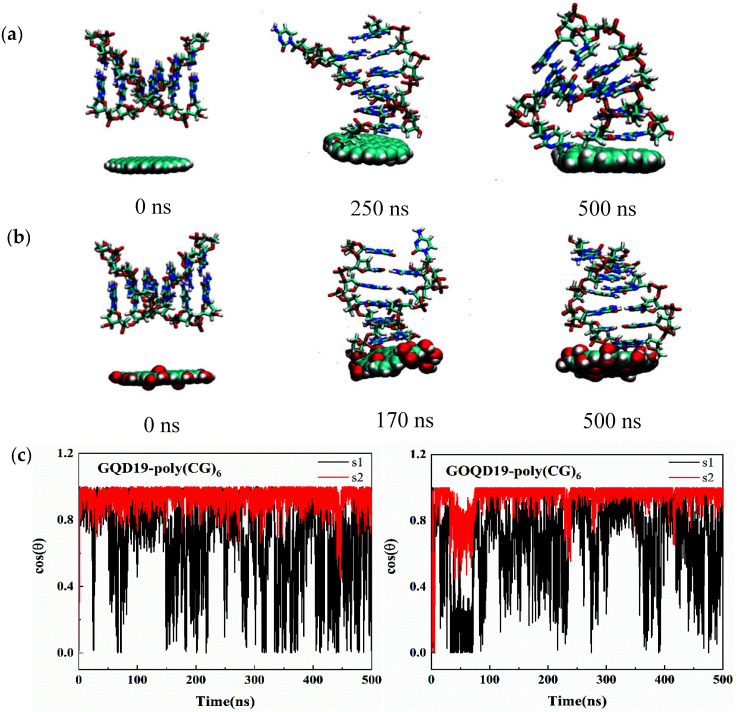
(**a**) Representative snapshots of dsDNA adsorption on (**a**) GQD19 and (**b**) GOQD19 surface. (**c**) The angle between the axis of the two single strands of dsDNA and the normal of the GQD19/GOQD19 surface during the simulation.

**Figure 5 materials-15-07435-f005:**
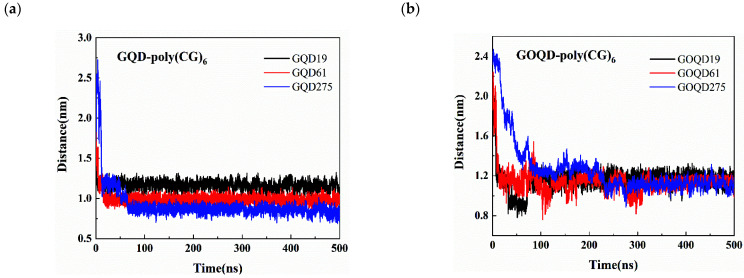
The COM distance between dsDNA and GQD/GOQD in *z*-direction during the simulation time: (**a**) GQD-poly(CG)_6_ and (**b**) GOQD-poly(CG)_6_.

**Figure 6 materials-15-07435-f006:**
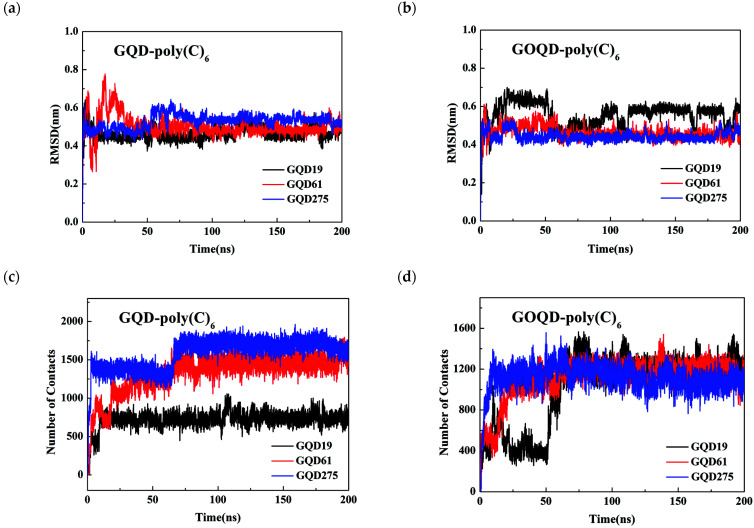
(**a**) The RMSD of poly(C)_6_ absorbed on (**a**) GQDs and (**b**) GOQDs surface with different sizes versus the simulation time. The number of contacts between poly(C)_6_ and (**c**) GQDs and (**d**) GOQDs with different sizes during the simulation.

**Figure 7 materials-15-07435-f007:**
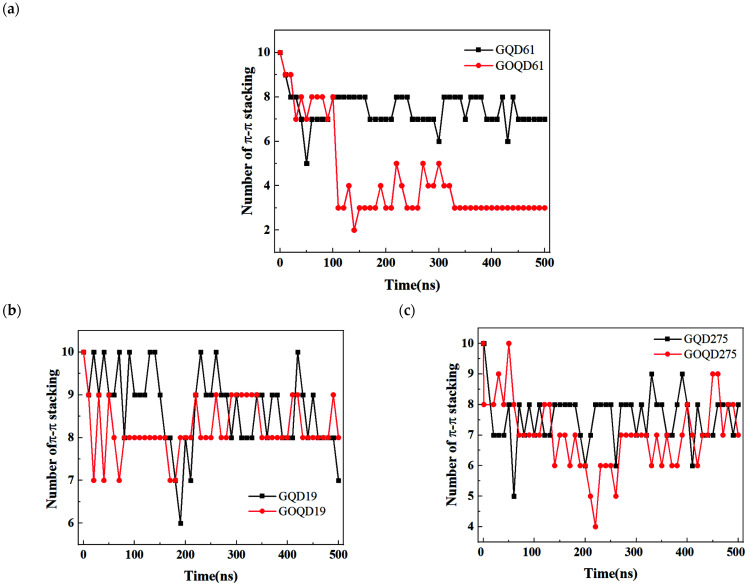
The π–π stacking number between dsDNA and GQD/GOQD with different sizes during the simulation: (**a**) GQD/GOQD 61; (**b**) GQD/GOQD 19; (**c**) GQD/GOQD 275.

**Figure 8 materials-15-07435-f008:**
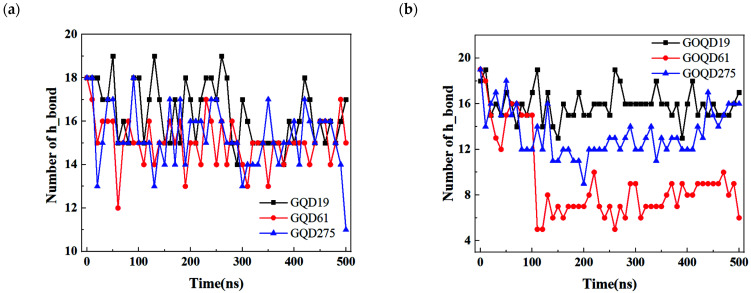
(**a**) The number of hydrogen bonds between the base pairs of dsDNA and GQDs with different sizes during the simulation. (**b**) The number of hydrogen bonds between the base pairs of dsDNA and GOQDs with different sizes during the simulation.

**Table 1 materials-15-07435-t001:** The details of all simulated systems in this work.

Name of System	GQD Type	DNA Type	Number of Na^+^	Number of Water Molecules	Simulation Time (ns)
GQD19-poly(C)_6_	GQD19	poly(C)_6_	5	6112	200
GQD61-poly(C)_6_	GQD61	poly(C)_6_	5	7403	200
GQD275-poly(C)_6_	GQD275	poly(C)_6_	5	12,047	200
GOQD19-poly(C)_6_	GOQD19	poly(C)_6_	5	6100	200
GOQD61-poly(C)_6_	GOQD61	poly(C)_6_	5	7387	200
GOQD275-poly(C)_6_	GOQD275	poly(C)_6_	5	11,941	200
GQD19-poly(G)_6_	GQD19	poly(G)_6_	5	6102	200
GQD61-poly(G)_6_	GQD61	poly(G)_6_	5	7397	200
GQD275-poly(G)_6_	GQD275	poly(G)_6_	5	12,029	200
GOQD19-poly(G)_6_	GOQD19	poly(G)_6_	5	6099	200
GOQD61-poly(G)_6_	GOQD61	poly(G)_6_	5	7384	200
GOQD275-poly(G)_6_	GOQD275	poly(G)_6_	5	11,925	200
GQD19-poly(CG)_6_	GQD19	poly(CG)_6_	10	6050	500
GQD61-poly(CG)_6_	GQD61	poly(CG)_6_	10	7330	500
GQD275-poly(CG)_6_	GQD275	poly(CG)_6_	10	11,959	500
GOQD19-poly(CG)_6_	GOQD19	poly(CG)_6_	10	6039	500
GOQD61-poly(CG)_6_	GOQD61	poly(CG)_6_	10	7310	500
GOQD275-poly(CG)_6_	GOQD275	poly(CG)_6_	10	11,891	500

## Data Availability

Not applicable.

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
