# Peer review of "Theoretical Evaluation of Potential Cytotoxicity of Graphene Quantum Dot to Adsorbed DNA"

_materials, 2022, doi:10.3390/ma15217435_

Round 1

Reviewer 1 Report

Dear Authors,

please find comments in the attached file.

Best Regards

Author Response

Response to Reviewers (materials-1975961)

Dear editor,

We thank the reviewers for the comments and suggestions that we found very helpful in improving this work and the resulting manuscript. The revision based on the reviewers’ comments has been completed and the changes are supplemented in the revised version of the manuscript. The point-by-point responses to each comment are included below. We hope that in its current stage, our manuscript may fulfill the high standards of Materials.

Review #1

In the present manuscript, the authors use molecular dynamics (MD) simulation to study the adsorption behavior and structural changes of single-stranded DNA (ssDNA) and double-stranded DNA (dsDNA) on the surfaces of GQDs of different sizes and oxidation. The work is very interesting and could contribute to a better understanding of the biocompatibility of GQDs and their potential applications in the biomedical field. For this reason, I recommend the paper for publication after revisions.

  1. Please unify the front page by using the materials template, the abstract must go on the first page.

Author reply: Thanks for the suggestion. We have already put the abstract on the first page.

  1. Several group acronyms were introduced in the abstract section. The reader cannot understand the meaning of GOQD, provide a previous definition.

Author reply: Thanks for your suggestive comment. We have provided a previous definition of graphene oxide quantum dots (GOQDs) in the abstract section. It was colored in red in revised manuscript.

  1. Rewrite references in the text as [6-7] and not [6,7,8].

Author reply: Thanks for the suggestion.  We have rewrote the format of the references.

  1. The introduction part can be reinforced with more discussion and more recent references (see DOI:10.1039/D2RA00494A, https://doi.org/10.3390/nano11112897, https://doi.org/10.3390/ma15165760)

Author reply: Thanks for the suggestion. These literatures could provide more comprehensive understanding on this topic. Agrawal et al. [R1] found that the modified GQDs have the efficient modification as well as higher stability, and enhanced biosensing efficiency and antibiofilm ability. Iannazzo et al. [R2] pointed that the GQDs-18-crown-6 composite exhibited ratiometric fluorescence emission behavior with the variation of K+ concentration, demonstrating its promising properties for the development of a selective fluorescent method for potassium determination. Ajgaonkar et al. [R3] developed a low-cost biocompatible and nitrogen-doped graphene quantum dots (NGQDs) sensor for pancreatic cancer miRNA (miRNA-132). This detection capability warrants the potential for ex vivo cancer miRNA detection with the advantages of being low-cost, simple, and noninvasive. We have supplemented these literatures in the revised manuscript. In addition, we have added several articles related to graphene and DNA. For example, graphene-based material is a strong absorber of lactate molecule, and this feature can be very useful in designing highly sensitive sensors [R4]. Graphene sheets also can be used to synthesize fast responding TSH nano-biosensors [R5]. DNA, being a genetic information carrier in living cells reveals tunable semiconducting response in the presence of external electric and magnetic fields, which is promising for molecular electronics. Khatir et al. [R6] revealed that the gold-DNA-gold structure is a promising magnetic sensor. These literatures were also supplemented in the introduction section in the revised manuscript. All changes were colored in red in revised manuscript.

  1. In figure 2bthe legend should be reduced because it almost overlaps the curves, making the figure confusing.

Author reply: Thanks for the comment. We have revised the Figure 2b accordingly in revised manuscript.

  1. Fix figure 4: the images are all overlapping

Author reply: Thanks for the suggestion. We have revised the Figure 4 accordingly in revised manuscript.

  1. Line 343: remove the Moreover, and move it to line 354, so that the sentence is not disconnected by figures 7 and 8 that follow.

Author reply: Thanks for the comment. We have done the revision on the basis of this suggestion.

  1. The conclusions are too general.

Author reply: Thanks for the suggestive comment.We have refined the conclusion as follows: “In addition to investigate the interaction between DNA and GQD with different sizes, we also focused on the effect of GOQD with various sizes on the adsorption of ssDNA as well as dsDNA, which was not addressed in previous studies ... In future work, we plan to study the effect of GOQD concentration, the degree of oxidation and DNA length on the interaction between GOQD and DNA. Moreover, the interaction mechanism between GQD/GOQD and protein, as well as the transmembrane process of GQD/GOQD are worth to investigated to fully understand the cytotoxicity of GQD/GOQD materials.” All changes were colored in red in revised manuscript.

  1. Manuscript needs moderate English spelling editing and grammar checks.

Author reply: We have tried our best to improve the English spelling and grammar. All changes were colored in red in revised manuscript.

These are all minor revisions, but they would improve the quality of this article.

Author reply: Thanks for your careful reading and very suggestive comments, and it could really improve the quality of this article.

Reviewer 2 Report

Dear Editor,

 I have read the manuscript entitled: “Theoretical evaluation of potential cytotoxicity of graphene quantum dot to adsorbed DNA” and I would like to address following suggestions to the authors:

1-Please add some lines to indicate the novelty of your study, compare the results with that of the literature and emphasize the novelty of this study.

2-For each research method, it is necessary to expand the discussion. Please add schematic diagram for this study.

3-Line 164: please mention in manuscript that temperature (K) is 300 and delete from table 1.

4-The introduction can be improved by providing a more critical discussion of recent related literature and use new papers. Discuss the shortcomings of previous work and the gaps and how this work intends to fill those gaps. For example, some papers related to graphen and DNA such as: Journal of Chemical and Petroleum Engineering, 55, 385-392, (2021); Superlattices and Microstructures, 145, 106603, (2020); Materials Science in Semiconductor Processing, 36, 134-139, (2015)  should be sited.

5-In the conclusion, the performance findings of the research should have been summarized the innovations and future scope of the work should be highlighted more.

Author Response

Response to Reviewers (materials-1975961)

Dear editor,

We thank the reviewers for the comments and suggestions that we found very helpful in improving this work and the resulting manuscript. The revision based on the reviewers’ comments has been completed and the changes are supplemented in the revised version of the manuscript. The point-by-point responses to each comment are included below. We hope that in its current stage, our manuscript may fulfill the high standards of Materials.

Review #2

  1. Please add some lines to indicate the novelty of your study, compare the results with that of the literature and emphasize the novelty of this study.

Author reply: Thank you very much for the suggestive comment. In the previous studies, the experiments have focused on the biocompatibility of GQDs and GOQDs. However, the atomic details of interaction mechanism between DNA and GQDs have not well understood. Several works focused on the adsorption and configuration changes of single-stranded (ssDNA) as well as dsDNA on GQDs, and they did not consider GOQDs with different sizes [R7, R8]. In this study, we used the standardized model to generate GOQDs with random distribution of oxidation groups of different sizes through home-based script, and both ssDNA and dsDNA were selected as model DNA to study the effects of different sizes and the oxidation of GQD on the structure and function of DNA by using MD simulation. These statements were supplemented at the end of introduction section,  and all changes were colored in red in revised manuscript.

  1. For each research method, it is necessary to expand the discussion. Please add schematic diagramfor this study.

Author reply: We have expanded our research method and supplemented these in the revised manuscript. Some of them are:

(1) CHARMM36 force field provides many parameters of biomolecules, such as proteins, lipids, nucleic acids, which is very suitable for simulation of DNA and nanomaterials [R9].

(2) Water molecule is represented by the TIP3P model [R10] , which is widely used in biological systems. The number of water molecules in each system is from 6,039 to 12,047 based on the size of systems.

 (3) LINCS algorithm is inherently stable, as the constraints themselves are reset instead of derivatives of the constraints, thereby eliminating drift. Although the derivation of the algorithm is presented in terms of matrices, no matrix matrix multiplications are needed and only the nonzero matrix elements have to be stored, making the method useful for very large molecules.

Moreover, we added the schematic diagram of the system from its initial conformation to conformations after energy minimization and MD simulation, as showed in Figure 1c. 

All changes were colored in red in revised manuscript.

Figure 1c. (c) Schematic diagram of the system from its initial conformation to conformations after energy minimization and MD simulation. Water is shown transparently.

  1. Line 164: please mention in manuscript that temperature (K) is 300 and delete from table 1.

Author reply: Thanks for the comment. The temperature we used in the simulation was 310K, which is the temperature of the physiological environment of the human body. We have removed the column of temperature(K) from the table 1 and supplemented another column containing the information of number of water molecules in each simulation system in table 1 in revised manuscript.

  1. The introduction can be improved by providing a more critical discussion of recent related literature and use new papers. Discuss the shortcomings of previous work and the gaps and how this work intends to fill those gaps. For example, some papers related to graphen and DNA such as: Journal of Chemical and Petroleum Engineering, 55, 385-392, (2021); Superlattices and Microstructures, 145, 106603, (2020); Materials Science in Semiconductor Processing, 36, 134-139, (2015)  should be sited.

Author reply: Thanks for the suggestive comments. We have revised the introduction section accordingly. Please see our reply to the 4th question of reviewer #1 and the section of introduction in revised manuscript (colored in red).

  1. In the conclusion, the performance findings of the research should have been summarized the innovations and future scope of the work should be highlighted more.

Author reply: Thanks for the suggestive comments. We have revised the conclusion section accordingly. Please see our reply to the 8th question of reviewer #1 and the section of conclusion in revised manuscript (colored in red).

Reviewer 3 Report

The manuscript entitled "Theoretical evaluation of potential cytotoxicity of graphene quantum dot to adsorbed DNA" written by Liang et al. reports results of molecular dynamics simulations of graphene sheets and modified graphene sheets (mainly oxidized) building adsorption complexes with ss and ds fragments of DNA. The main research question is cytotoxicity, which could be evaluated via investigation of the DNA secondary structure loss during and after adorption processes onto the surface of the modelled dots. 

It has been shown that ssDNA lies flat, while dsDNA is standing upright. Also, the authors conclude about the influence of the dot size and its oxidation on the structural damage of the DNA fragment.

This manuscript is probably publishable, but a few points should be taken into account prior the final decision is made.

1. The athors should clearly correlate the size of the dot and the structural damage of DNA fragment caused by the adsorption. It would be also helpful to explain which sizes and types of the dots are studied in experimental works from literature.

2. In a living cell everything happens in watery environment. From the description of the model it is absolytely unclear how many water molecules, using which model, etc. have been included in the simulation boxes.

3. Graphene and Graphene oxides could make stacks in water. What would the authors expect in this case? can DNA make a shell around such stacks? Why the authors have not considered the interaction of DNA with the edges of the dots? This is very welcome to ensure the completness of the presented research.

4. Please correct the following Figure  (some text is cut): 4

5. Please differentiate between the published work 52 and presented in this manuscript results.

Author Response

Response to Reviewers (materials-1975961)

Dear editor,

We thank the reviewers for the comments and suggestions that we found very helpful in improving this work and the resulting manuscript. The revision based on the reviewers’ comments has been completed and the changes are supplemented in the revised version of the manuscript. The point-by-point responses to each comment are included below. We hope that in its current stage, our manuscript may fulfill the high standards of Materials.

Review #3

The manuscript entitled "Theoretical evaluation of potential cytotoxicity of graphene quantum dot to adsorbed DNA" written by Liang et al. reports results of molecular dynamics simulations of graphene sheets and modified graphene sheets (mainly oxidized) building adsorption complexes with ss and ds fragments of DNA. The main research question is cytotoxicity, which could be evaluated via investigation of the DNA secondary structure loss during and after adorption processes onto the surface of the modelled dots. 

It has been shown that ssDNA lies flat, while dsDNA is standing upright. Also, the authors conclude about the influence of the dot size and its oxidation on the structural damage of the DNA fragment.

This manuscript is probably publishable, but a few points should be taken into account prior the final decision is made.

  1. The authors should clearly correlate the size of the dot and the structural damage of DNA fragment caused by the adsorption. It would be also helpful to explain which sizes and types of the dots are studied in experimental works from literature.

Author reply: Thanks for the suggestive comment. We revised relevant statement and emphasized the influence of size on DNA structure in lines 389 to 395 in revised manuscript: In this study, we found that the adsorption of ssDNA and dsDNA on the surface of GQD has stronger size effect. Comparing to GOQD, the larger the size of GQD, the more obvious the damage to DNA structure. In particular, GQD275 caused the most obvious damage to DNA structure. GQD275 not only lead to most atomic contacts with poly(C)6, poly(G)6 and poly(CG)6, but also had the largest changes in RMSD values of them on its surface. And the destruction of ssDNA and dsDNA structure on GQD surface is higher than that on GOQD surface.

These results are consistent with the results of Jeong et al.[R8]. By means of experiments and MD simulation, Jeong et al. found that compared with GOQD, GQD has stronger adsorption to ssDNA and greater damage to the structure of ssDNA. However, they didn't show the average size of GQD and GOQD. Wang et al. [R11] found that GQD can induce DNA damage in NIH-3T3 cells, which lead to cytotoxicity. The average lateral size of GQD used by Wang et al. in their experiments is 40 nm. In our study, the lateral sizes of GQD/GOQD are about 1- 4 nm, due to the limitation of building large box in MD simulation. However, our result shows that the size effect of GQD is relatively strong and GQD275 (~ 4 nm in size) can cause more obvious structure damage of DNA, the GQD with size of ~40 nm may cause more stronger deformation of DNA structure and lead to cytotoxicity, as Wang et al. found in their experiment.

  1. In a living cell everything happens in watery environment. From the description of the model it is absolutely unclear how many water molecules, using which model, etc. have been included in the simulation boxes.

Author reply: Thanks for your suggestive comment. Water molecule in this study is represented by the TIP3P model [R10], which is widely used in biological systems. The number of water molecules in each system is from 6,039 to 12,047 based on the size of systems. We supplemented another column containing the information of number of water molecules in each simulation system in table 1. All changes were colored in red in revised manuscript.

  1. Graphene and Graphene oxides could make stacks in water. What would the authors expect in this case? can DNA make a shell around such stacks? Why the authors have not considered the interaction of DNA with the edges of the dots? This is very welcome to ensure the completness of the presented research.

Author reply: Thanks for your suggestive comments. Solanky et al. [R12] tested the inherent hydrophobic behavior of a small graphene in water droplet by the means of MD simulations. The analysis has been extended to multiple graphene flakes in water and their respective size dependent responses to water droplet. Graphene retreats from water droplet to encapsulate it from the surface. This response was highly dependent upon graphene size with respect to water content. Rubim et al. [R13] described a procedure to obtain GO dispersions in water at high concentrations, and these highly dehydrated dispersions being in addition fully redispersible by dilution, thereby evidencing the relevance of both electrostatic and steric (Helfrich) interactions in stabilising aqueous lamellar stacks of GO sheets. Our previous work considered DNA adsorption at the edges of GQD [R7]. We found that only GQDs with small size (GQD7) could enter into the interior of the DNA fragment and break the hydrogen bonds of the DNA fragment. The large-size GQDs (GQD61) tend to aggregate into a cluster and adsorb on the DNA fragment, as shown in Figure 1. In addition, GQD61 tends to insert into the terminal of poly(A−T)20, and it largely breaks the hydrogen bond between the A−T bases in a high concentration of GQD61, as shown in Figure 2. In this work, we focused on the size effects and oxidation of GQDs on DNA adsorption. In future work, we plan to study the effect of GOQD concentration, the degree of oxidation and DNA length on the interaction between GOQD and DNA. Moreover, the interaction mechanism between GQD/GOQD and protein, as well as the transmembrane process of GQD/GOQD are worth to investigated to fully understand the cyto-toxicity of GQD/GOQD materials.

Figure 1. Absorption of GQDs with different sizes on poly(A–T)20 at the simulation time of 100 ns: (a) GQD7, (b) GQD19, and (c) GQD61. (green: dA, red:dT, blue: GQDs).

Figure 2. Snapshot of poly(A–T)20 in the condition of different concentrations of GQD61 at the end of simulation: (a) GQD61-L, (b) GQD61-M, and (c) GQD61-H.

  1. Please correct the following Figure  (some text is cut): 4

Author reply: Thanks for the suggestion. As the reviewer mentioned, we have revised the Figure 4 accordingly.

  1. Please differentiate between the published work 52 and presented in this manuscript results.

Author reply: Thanks for your suggestive comment. In Published work 52, the authors reported the study on cytotoxicity and genotoxicity of GQDs to fibroblast cell lines (NIH-3T3 cells). The NIH-3T3 cells treated with GQDs at dosages over 50 μg/mL showed no significant cytotoxicity. However, the GQD-treated NIH-3T3 cells exhibited an increased expression of proteins (p53, Rad 51, and OGG1) related to DNA damage compared with untreated cells, indicating the DNA damage caused by GQDs. The GQD-induced release of reactive oxygen species (ROS) was demonstrated to be responsible for the observed DNA damage. Through experiments, they found that GQDs can cause DNA damage, but there was no reference to the sizes of GQDs, nor to the specific type of GOQDs. In our study, MD simulation was used to explore the adsorption behavior and dynamics of ssDNA/dsDNA on GQD and GOQD with different sizes. The effects of size and oxidation of GQD on the mechanism of DNA adsorption were investigated. During the adsorption process, ssDNA tends to adsorb on the surfaces of GQD/GOQD in a parallel manner, while dsDNA tends to adsorb on the surfaces of GQD/GOQD in a nearly perpendicular orientation. GQD has greater influence on the structure change of DNA than GOQD, and the size effect of GQD on DNA structure change is more obvious than that of GOQD.

Round 2

Reviewer 3 Report

None. I recommend accepting this manuscript